# Comparative Study of the Preparation of High-Molecular-Weight Fibroin by Degumming Silk with Several Neutral Proteases

**DOI:** 10.3390/polym15163383

**Published:** 2023-08-12

**Authors:** Xueping Liu, Qian Huang, Peng Pan, Mengqi Fang, Yadong Zhang, Shanlong Yang, Mingzhong Li, Yu Liu

**Affiliations:** National Engineering Laboratory for Modern Silk, College of Textile and Clothing Engineering, Soochow University, Suzhou 215123, China; 18862107248@163.com (X.L.); 15895571967@139.com (Q.H.); ppanpanpeng@stu.suda.edu.cn (P.P.); 20214215051@stu.suda.edu.cn (M.F.); 20215215066@stu.suda.edu.cn (Y.Z.); yangsl886886@163.com (S.Y.)

**Keywords:** silk fibroin, degumming, neutral protease, molecular weight

## Abstract

Removing sericin from the periphery of silk without damage to silk fibroin (SF) to obtain high-molecular-weight SF is a major challenge in the field of SF-based biomaterials. In this study, four neutral proteases, subtilisin, trypsin, bromelain and papain, were used to degum silk, and the degumming efficiency of the proteases and their influence on the molecular weight (MW) of regenerated silk fibroin were studied. The results indicated that all four neutral proteases could remove sericin from silk almost completely, and they caused less damage to SF fibers than Na_2_CO_3_ degumming did. The degumming efficiency of trypsin and papain was strong, but they caused relatively high damage to SF, whereas bromelain caused the least damage. The results of sodium dodecyl sulfate–polyacrylamide gel electrophoresis, gel permeation chromatography and shear viscosity showed that the MWs of regenerated SF derived from neutral protease degumming were significantly higher than that of SF derived from Na_2_CO_3_ degumming. The MW of regenerated SF derived from bromelain degumming was the highest, while the MWs of regenerated SF derived from papain and trypsin degumming were relatively low. This study provides an efficient and environmentally friendly biological degumming method for obtaining high-molecular-weight silk fibroin.

## 1. Introduction

Silk fibroin (SF) has broad application prospects in the field of tissue engineering due to its good cytocompatibility, biodegradability and low immunogenicity [1]. The mechanical properties, repair cycle, and tissue growth rate of different tissues (such as bone, tendon, ligament, and blood vessel) vary greatly, resulting in varying requirements for the mechanical strength and biodegradation rate of SF-based tissue engineering scaffolds. The molecular weight (MW) of SF is the key factor in determining the mechanical properties and biodegradation rate of regenerated silk fibroin materials [2]. An effective way to fabricate SF materials to meet different biomedical needs is by controlling the MW of silk fibroin to regulate the structure and properties of regenerated SF materials. The MW of natural silk fibroin is approximately 2286 kDa, but degumming and dissolution processes significantly reduce the MW of regenerated SF [3,4]. Obtaining high-molecular-weight regenerated SF is one of the challenges in the field of SF-based biomedical materials.

Degumming is the first step in obtaining regenerated silk fibroin. The degumming effect and the degree of damage to silk fibroin fibers directly affect the molecular weight of silk fibroin [5,6,7]. Based on the differences in amino acid composition and molecular conformation between silk fibroin and sericin, various degumming methods such as using alkaline reagents, acidic reagents, a high temperature and high pressure, soap and enzymes have been developed. Alkaline reagents have a strong dissolution effect on sericin and high degumming efficiency, but they cause significant damage to silk fibroin fibers, resulting in a significant reduction in the MW of silk fibroin [8,9]. Degumming conditions can be optimized by controlling the temperature and alkali concentration, and complete degumming while avoiding excessive damage of SF is feasible at a lower temperature and lower alkali concentration [10]. The effect of organic acids on silk is milder than that of alkaline reagents, but they easily adsorb on silk fibroin fibers and are extremely difficult to remove [11,12]. Inorganic acids have a stronger effect on sericin than organic acids do, but the degumming effect is difficult to control, and they also cause damage to silk fibroin fibers [11,13]. High-temperature and high-pressure degumming conditions are harsh, and only the outer layer of sericin can be removed [14]. Soap degumming requires high soap quality and water quality as well as involving high water consumption and high costs [15,16]. Protease, as an environmentally friendly biocatalyst, has mild degumming conditions (in terms of temperature and pH value) and low energy consumption. Enzymatic degumming has good specificity and high catalytic efficiency, and it causes less damage to silk fibroin fibers during degumming [17]. Moreover, the mixed waste liquid of sericin and enzyme generated during the degumming process is biodegradable, resulting in low pollution [18,19]. Therefore, enzyme treatment is an ideal degumming process for silk degumming.

Compared to alkaline proteases, neutral proteases such as subtilisin, trypsin, bromelain, and papain all show the advantages of having mild degumming conditions and causing less damage to silk fibroin fibers, indicating their potential to be developed and applied in practical production [20,21,22]. Both subtilisin and trypsin are serine proteases. The properties of subtilisin are affected by the type of bacteria. The subtilisin obtained via the submerged fermentation of *Bacillus subtilis* 1398 has been used for the degumming of silk in the early stage, and it can catalyze the hydrolysis of protein into small-molecule peptides under neutral, weakly acidic or weakly alkaline conditions [23]. The main cleavage site of trypsin is a peptide bond formed by the carboxyl groups of basic amino acids (arginine and lysine) and the amino groups of other amino acids, and the product is a peptide whose carboxyl terminus is lysine or arginine. The contents of arginine and lysine in sericin are significantly higher than those in silk fibroin, so trypsin has a certain degree of specificity to sericin [24,25]. The active sites of bromelain and papain contain cysteine, so they are sulfhydryl proteases. Bromelain preferentially hydrolyzes the carboxyl terminus of arginine in proteins and peptides. Bromelain can be divided into stem bromelain and fruit bromelain according to the different extraction parts. In this study, stem bromelain was selected [26]. Papain mainly acts on the peptide bond between an arginine or lysine residue and another amino acid residue, so both papain and bromelain are specific to sericin [27,28,29].

The aim of this study is to obtain a mild degumming method with neutral protease that can not only effectively remove sericin from the outer layer of silk but also prevent the excessive destruction of silk fibroin, limiting the decrease in the MW of silk fibroin. For this purpose, subtilisin, trypsin, bromelain and papain were used to degum *Bombyx mori* silk at the optimal temperature and pH value of these four neutral proteases, and the sodium carbonate degumming method was used as a control. The degumming efficiency of each neutral protease was investigated with weight degumming ratio, scanning electron microscopy (SEM) and amino acid analysis, and the effect of the proteases on the thermal stability of degummed silk fibers was examined by thermogravimetric analysis. Then, the degummed silk fibers were dissolved, dialyzed and purified to obtain regenerated silk fibroin. The molecular weight of regenerated SF was characterized by sodium dodecyl sulfate–polyacrylamide gel electrophoresis (SDS–PAGE), gel permeation chromatography (GPC) and shear viscosity, and the effect of the degumming of different neutral proteases on the molecular weight of regenerated SF was studied.

## 2. Materials and Methods

### 2.1. Silk Degumming with Different Neutral Proteases

Subtilisin (EC 3.4.4.16, 6  ×  10^4^ U/g), bromelain (EC 3.4.22.4, 5  ×  10^5^ U/g) and papain (EC 3.4.22.2, 8  ×  10^5^ U/g) were all purchased from China, Beijing Solebel Co., Ltd. Trypsin (EC 3.4.21.4, 5  ×  10^4^ U/g) and Na_2_CO_3_ were purchased from Shanghai, China, Sinopharm Chemical Reagent Co., Ltd. Eight-gram samples of *Bombyx mori* raw silk (SOHO Biomaterials Science and Technology Co., Ltd., Suzhou, China) were placed in 400 mL of solutions containing subtilisin, trypsin, bromelain and papain. According to the protease commodity information and based on previous research, optimal degumming conditions were used for each of the four enzymes [23,30,31], as shown in Table 1. The pH values were adjusted with sodium dihydrogen phosphate–citric acid buffer. The degummed silks were immediately placed into boiling water to inactivate the enzymes and then thoroughly washed with distilled water. The above steps were repeated once. The traditional Na_2_CO_3_-degumming method was used as a control. In brief, 8 g of raw silk was boiled three times for 30 min each in 400 mL of 0.5 g/L Na_2_CO_3_ aqueous solution and then thoroughly rinsed with deionized water. Next, all degummed silk fibers were air-dried in an oven at 60 ± 2 °C. The degummed silk fibers were marked as subtilisin-silk, trypsin-silk, bromelain-silk, papain-silk and Na_2_CO_3_-silk.

### 2.2. Degumming Ratio

The degumming ratio was assessed via the mass lost during the degumming process. The conditioned weights, m_1_(g) and m_2_(g), of silk before and after degumming were calculated according to the dry weight of silk before and after degumming. The degumming ratio P (%) was calculated according to Equation (1):(1)$$P\left(\%\right)=\frac{m_{1}(g)\;-m_{2}\left(g\right)}{m_{1}\left(g\right)}\times 100$$

### 2.3. Morphological Characterization of Degummed Silk Fibers

The raw silk and degummed silk fibers were sprayed with gold for 90 s, and then the surface morphologies were observed using a Hitachi S-4800 scanning electron microscope (Hitachi, Japan).

### 2.4. Amino Acid Analysis

An amino acid analysis was carried out on Hitachi L-8900 Amino Acid Analyzer (Hitachi, Ltd., Tokyo, Japan). The raw silk and degummed silk fibers were hydrolyzed in 6 mol/L of HCl for 24 h at 110 °C. After removing the HCl, the concentration of the residues was diluted to 200 ng/mL with 0.02 mol/L HCl and they were filtered with Millipore 0.22-μm syringe filters (Milford, CT, USA).

The sericin content in degummed silks can be quantitatively characterized according to the difference in amino acid content between sericin and silk fibroin [32]. The content of aspartic acid (Asp) in sericin is higher than that in silk fibroin, while the content of alanine (Ala) in sericin is lower than that in silk fibroin. Thus, the sericin containing ratio y (%) of degummed silks was calculated according to Equations (2) and (3):(2)$$x=\frac{\mathrm{Asp}(\mathrm{mol}\%)}{\mathrm{Ala}(\mathrm{mol}\%)}\times 100$$
(3)$$y(\%)=2.6913x\;-14.2885\;(R^{2}=0.9958)$$

### 2.5. Thermal Analysis of Degummed Silk Fibers

The raw silk and degummed silk fibers were cut into microparticles. Thermal analysis was performed with Perkin-Elmer Diamond Thermal Analyzer (Waltham, MA, USA). The heating rate was 10 °C/min under a flow of N_2_, and the test range was 40–600 °C.

### 2.6. Preparation of Regenerated SF Solution

One-gram samples of the degummed silk fibers, subtilisin-silk, trypsin-silk, bromelain-silk, papain-silk and Na_2_CO_3_-silk, were dissolved in 30 mL of 9.3 mol/L LiBr (Hefei, China) aqueous solution at 60 ± 2 °C for 1 h. After cooling, the mixtures were dialyzed against deionized water at 4 °C for 3 days with a dialysis bag (MWCO 8–14 kDa) and were filtered to obtain the SF solution. The resulting regenerated silk fibroins were labeled subtilisin-SF, trypsin-SF, bromelain-SF, papain-SF and Na_2_CO_3_-SF.

### 2.7. Characterization of the Molecular Weight Distribution of Regenerated Silk Fibroin

Sodium dodecyl sulfate–polyacrylamide gel electrophoresis (SDS–PAGE): The MW distribution of regenerated SF was examined using SDS–PAGE. The concentration of the stacking gel was 5%, and the concentration of the separating gel was 6% and 10%. Prestained proteins were used as MW markers (43–315 and 10–200 kDa, Thermo Fisher Scientific, Waltham, MA, USA). The concentration of added SF solution was adjusted to 4 mg/mL. After electrophoresis, the gels were stained with Coomassie brilliant blue R-250 (Aladdin, Shanghai, China). The gel images were analyzed using ImageJ 1.51J8 software, and the MW of the samples was calculated indirectly by calculating the magnitude of the gray value of each band [33].

Gel permeation chromatography (GPC): 0.1% (*w*/*v*) SF solutions containing 4 mol/L urea aqueous solution were prepared and filtered with a 0.45 μm filter membrane and then tested using a rapid purification liquid chromatography system (AKTA purifiers, GE Healthcare, Chicago, IL, USA). A 4 mol/L urea aqueous solution was used as the mobile phase. A 100 μL aliquot of solution was injected and the mobile phase (4 mol/L urea) of 4 column volumes was pumped at a constant flow rate of 0.5 mL/min. The protein concentration was observed via UV absorbance at 280 nm.

### 2.8. Measurement for Shear Viscosity of Regenerated SF Solution

The steady-state flow curve of the shear viscosity of the regenerated SF solution with a shear rate was determined using a rheometer (AR-2000, TA Instruments, New Castle, DE, USA) with a 40 mm/1° tapered plate at 25 °C. A shear rate range of 0.01–10,000 s^−1^ was applied.

### 2.9. Statistical Analysis

All experiments were performed in triplicate, and the results are expressed as the mean ± SD of triplicate assays. The SPSS 17.0 statistical software package (IBM) was used for data processing. Statistical analyses were performed using an unpaired Student′s *t* test. Differences between groups were considered statistically significant when the p value was less than 0.05.

## 3. Results

### 3.1. Degumming Ratio and Morphological Characterization of Degummed Silk Fibers

Figure 1 shows the degumming ratio of different degumming methods and the surface morphologies of degummed silk fibers. As shown in Figure 1A, the degumming ratio of the traditional Na_2_CO_3_ method was 22.93 ± 0.47%, and the degumming ratios of subtilisin, trypsin, bromelain and papain were 22.3 ± 1.01%, 24.53 ± 1.41%, 21.76 ± 0.2% and 23.85 ± 0.52%, respectively. The results suggested that there was no significant difference between the degumming ratios of different neutral proteases and that of Na_2_CO_3_. The degumming ratios of trypsin and papain were relatively high and significantly higher than that of bromelain, indicating that the degumming was relatively sufficient. The SEM photograph of silk fibers degummed with different reagents is shown in Figure 1B. The surface of the raw silk was rough, and many filaments were tangled with each other. Sericin adhered to the surface of the filaments with an uneven distribution (Figure 1B(f)), while the fiber bundles of degummed silk were clearly visible (Figure 1B(a–e)). The surface of Na_2_CO_3_-silk was smooth, and there were deep grooves along the fiber axis (Figure 1B(e)). The surface of silk fibers degummed by different neutral proteases had a small amount of granular attachments, subtilisin-silk, bromelain-silk and papain-silk had shallow vertical stripes along the fiber axis (Figure 1B(a,c,d)), and trypsin-silk had grooves along the fiber axis (Figure 1B(b)). In addition, the diameters of degummed silk fibers were smaller than that of raw silk (Appendix A). The results indicated that the neutral proteases could almost completely remove sericin, but they also caused some damage to SF fibers and trypsin caused relatively significant damage to some SF fibers.

### 3.2. Amino Acid Analysis and Sericin Containing Ratio

Fibroin and sericin are clearly different in amino acid composition and content. Therefore, the amino acid composition of silk before and after degumming can reflect the degumming degree to some extent. The contents of amino acids in the raw silk and degummed silk fibers are listed in Table 2. As shown in Table 2, there was a great difference in the amino acid composition in silk fibers before and after degumming, in which the contents of Asp, Thr, Ser and Glu were obviously decreased, while the contents of Gly and Ala were obviously increased. According to the contents of Asp and Ala in degummed silk, the sericin containing ratio of degummed silk fibers can be calculated quantitatively [32], and the result is shown in Figure 2. The sericin containing ratio of raw silk was 24.71 ± 0.1%. However, the sericin containing ratios of silk fibers degummed with subtilisin, trypsin, bromelain and papain were 0.71 ± 0.08%, 0.64 ± 0.04%, 0.76 ± 0.09% and 0.67 ± 0.14%, respectively, which were slightly higher than that of silk fibers degummed with Na_2_CO_3_ (0.33 ± 0.07%). The results showed that neutral proteases could remove sericin to varying degrees.

### 3.3. Thermal Analysis

Figure 3 shows the derivative thermogravimetric (DTG) curves of silk fibers degummed by different neutral proteases and sodium carbonate. The maximum thermal decomposition rate temperature of raw silk was approximately 324.7 °C. The maximum thermal decomposition rate temperatures of subtilisin-silk, trypsin-silk, bromelain-silk and papain-silk were 326.7, 321.9, 331.0 and 327.5 °C, respectively, which were all higher than that of Na_2_CO_3_-silk (320.9 °C), indicating that the thermal stabilities of silk degummed by neutral proteases were better than that of silk degummed by Na_2_CO_3_, and bromelain-silk had the best thermal stability.

### 3.4. Characterization of the Molecular Weight Distribution of Regenerated Silk Fibroin

The SDS–PAGE results in Figure 4 show the electrophoretic bands of the regenerated SF solutions derived from subtilisin, trypsin, bromelain, papain and Na_2_CO_3_ degumming. In the gel with 6% separating gel, the regenerated SF derived from neutral protease degumming had higher MW bands distributed above 180 kDa (Figure 4A(a–d)), and there were clear bands at 30 and 25 kDa in the gel with 10% separating gel (Figure 4B(a–d)). However, Na_2_CO_3_-SF had a broad MW distribution in the lower MW ranges, with sequential bands from 95 to 250 kDa (Figure 4A(e)), and there were no obvious bands at 30 and 25 kDa (Figure 4B(e)). The results indicated that degumming by different neutral proteases caused less damage to the SF segment, and the molecular weights of subtilisin-SF, trypsin-SF, bromelain-SF and papain-SF were obviously higher than that of Na_2_CO_3_-SF.

A semiquantitative analysis of the MW distribution was performed by measuring the gray intensity values of different MW regions. As shown in Figure 5, subtilisin-SF, trypsin-SF, bromelain-SF and papain-SF had more components in the high-MW region than Na_2_CO_3_-SF did. In the gel with 6% separating gel, the average MWs of subtilisin-SF, trypsin-SF, bromelain-SF, papain-SF and Na_2_CO_3_-SF were 186.5 ± 7.7, 179.5 ± 14.8, 186.5 ± 6.3, 172.5 ± 2.1 and 156.5 ± 0.7 kDa, respectively (Figure 5A). In the gel with 10% separating gel, the average MWs of subtilisin-SF, trypsin-SF, bromelain-SF, papain-SF and Na_2_CO_3_-SF were 166.5 ± 12, 168 ± 11.3, 170 ± 11.3, 163.5 ± 3.5 and 133 ± 5.6 kDa, respectively (Figure 5B). The results indicated that the average MWs of the regenerated SF derived from degumming with the four neutral proteases were significantly greater than that of Na_2_CO_3_, and the average MWs of subtilisin-SF, trypsin-SF, bromelain-SF showed no obvious difference, while the average MW of papain-SF was relatively low.

Figure 6 shows the chromatographic leaching curves of the regenerated SF solution derived from degumming with different proteases and Na_2_CO_3_. Subtilisin-SF, trypsin-SF, bromelain-SF and papain-SF all began to flow out at approximately 7.67 mL (Figure 6(a–d)), while Na_2_CO_3_-SF began to flow out at approximately 8.06 mL (Figure 6(e)), indicating that the highest molecular weights of silk fibroin derived from neutral protease degumming were basically the same, and they were higher than that of Na_2_CO_3_-SF. The elution peaks of subtilisin-SF and bromelain-SF were at 8.51 mL and 8.50 mL (Figure 6(a,c)), respectively, and the peak shape of bromelain-SF was high and narrow (Figure 6(c)), indicating that the highest MWs of subtilisin-SF and bromelain-SF were close, but the MW distributions of bromelain-SF were concentrated, and bromelain-SF contained more high-molecular-weight segments, while the MW distributions of subtilisin-SF were relatively dispersed. The elution peak of papain-SF (8.57 mL) appeared earlier than that of trypsin-SF (8.67 mL), indicating that the highest MW of papain-SF was higher than that of trypsin-SF (Figure 6(d,c)). However, the elution peak of the gel chromatography curve of Na_2_CO_3_-SF was at 8.90 mL, and the peak shape was short and wide (Figure 6(e)). The results suggested that the MW distribution of the silk fibroin derived from Na_2_CO_3_ degumming was wide, mainly in the lower-molecular-weight region.

### 3.5. Viscosity of Regenerated SF Solution

The viscosity of the solution is another important indicator of the molecular weight of silk fibroin, because the viscosity of the SF solution increases with increasing molecular weight [34]. As shown in Figure 7, when the shear rate was 0.01–0.1 rad/s, the shear viscosities of different regenerated SF solutions had no obvious changes with the increasing shear rate, and the shear viscosity of bromelain-SF was the largest, while that of subtilisin-SF and trypsin-SF showed no obvious difference, which were between that of bromelain-SF and papain-SF. The shear viscosities of subtilisin-SF, trypsin-SF, bromelain-SF and papain-SF were all higher than that of Na_2_CO_3_-SF. When the shear rate was 0.1–1000 rad/s, the viscosities of the silk fibroin solutions decreased rapidly with increasing shear speed. When the shear rate was greater than 1000 rad/s, the shear viscosities of different silk fibroin solutions remained basically unchanged and there was no significant difference. The results suggested that compared with Na_2_CO_3_ degumming, the degree of hydrolysis of silk fibroin via protease degumming was lower, and the MW of the resulting silk fibroin was higher. The MW of SF derived from bromelain degumming was the highest, and the MW of SF derived from papain degumming was the lowest. There was no obvious difference between the MWs of SF derived from subtilisin and trypsin degumming, which were between those of bromelain and papain.

## 4. Discussion

The molecular weight of silk fibroin has a significant effect on the structure and properties of regenerated silk fibroin materials, which is the key factor in determining the mechanical properties and biodegradation rate of materials. Degumming is a key step in purifying silk fibroin. This process will destroy the amorphous region structure of SF and hydrolyze SF to varying degrees, resulting in a decrease in the MW of regenerated silk fibroin, which has a significant impact on the MW distribution of SF, ultimately affecting the performance of silk fibroin materials in biomedical applications such as tissue engineering [35]. In addition, effective degumming is essential for the biocompatibility and immunogenicity of SF materials; therefore, it is necessary to completely remove sericin while preserving SF integrity [36]. Some neutral proteases can act on specific amino acid sequence sites on the sericin main chain to catalyze the hydrolysis of peptide bonds, thus removing sericin from silk and simultaneously reducing the damage to SF fibers, thereby reducing the impact on the MW of regenerated silk fibroin [7,21]. In this study, four neutral proteases, subtilisin, trypsin, bromelain and papain, were used to degum silk, and the degree of damage caused by these four neutral proteases to silk fibroin fibers and their influence on the molecular weight of regenerated silk fibroin were studied.

The degumming efficiency of neutral proteases was characterized using the degumming rate, SEM and amino acid analysis. The degumming rates of subtilisin, trypsin, bromelain and papain were not significantly different from that of Na_2_CO_3_ (Figure 1A), and the degumming ratios of trypsin and papain were higher than that of Na_2_CO_3_, indicating that degumming was comparably sufficient. However, the sericin containing ratios of silk fibers degummed with proteases were higher than those of Na_2_CO_3_-silk (Figure 2), which might be because different degumming treatments performed different degrees of hydrolysis on the silk fibroin fibers, resulting in different amino acid contents in the silk fibroin fibers. The SEM images in Figure 1B show that the different neutral proteases could almost completely remove sericin while causing some damage to SF, but compared to Na_2_CO_3_ degumming, they caused less damage to SF fibers. The deep grooves along the fiber axis on Na_2_CO_3_-silk might have been caused by the peeling of some fibers from the surface of the SF fibers during the degumming process (Figure 1B(e)). The vertical stripes along the fiber axis on bromelain-silk, subtilisin-silk and papain-silk were caused by the uneven action of proteases on silk and somewhat excessive degumming (Figure 1B(a,c,d)). Trypsin caused significant damage to SF. The surface of trypsin-silk was rough and grooved, which might have been due to the partial hydrolysis of SF fiber by trypsin, and fibrils peeled off from the fiber surface, resulting in significant mass loss and a high degumming rate (Figure 1B(b)).

The maximum thermal decomposition rate temperatures of silk fibers degummed by neutral proteases were higher than those of SF fibers degummed by Na_2_CO_3_, because Na_2_CO_3_ caused significant structural damage to SF during degumming (Figure 3). The maximum thermal decomposition rate temperatures increased in the following order: trypsin-silk, subtilisin-silk, papain-silk, and bromelain-silk. Bromelain-silk had the highest maximum thermal decomposition rate temperature, indicating that the damage to silk fibroin caused by bromelain degumming was the lowest, and the obtained SF fiber had the best thermal stability.

*Bombyx mori* silk fibroin is composed of heavy chains (350 kDa), light chains (25.8 kDa) and glycoprotein P25 chains (23.55 kDa). Heavy chains and light chains are linked by disulfide bonds, and P25 chains are bound to heavy chains by noncovalent bonds [37]. SDS–PAGE results showed that regenerated SF derived from subtilisin, trypsin, bromelain and papain degumming all had continuous bands above 180 kDa (Figure 4A), indicating that all silk fibroin derived from the proteases degumming contained high-MW segments. In addition, the bands at 30 kDa and 25 kDa were clearly visible (Figure 4B), corresponding to the light chains and P25 chains in silk fibroin, respectively, indicating that proteases broke disulfide bonds and hydrophobic bonds during degumming [38]. However, Na_2_CO_3_-SF shows a continuous band at 95–250 kDa (Figure 4A) but no obvious bands at 30 and 25 kDa (Figure 4B). This was because silk fibroin was degraded into lower-molecular-weight fragments during the Na_2_CO_3_ degumming process, and the light chains and P25 chains were also partially degraded [39]. ImageJ software was used to analyze the gray intensity values of SDS–PAGE images to indirectly calculate the MW of the samples (Figure 5). In different separation gels, there was a certain difference in the average MW of regenerated SF derived from the same protease degumming procedure, which might have been caused by the difference in prestained proteins used in the corresponding separation gels. The results showed that the average MWs of regenerated SF derived from degumming with the four neutral proteases were higher than those of SF derived from Na_2_CO_3_ degumming. There was no obvious difference between the average MWs of regenerated SF derived from subtilisin, trypsin and bromelain degumming, while the average MW of regenerated SF derived from papain degumming was relatively low. In addition, the results of GPC showed that the highest MW of regenerated SF derived from neutral protease degumming was higher than that of Na_2_CO_3_-SF. The highest MWs decreased in the following order: bromelain-SF, subtilisin-SF, papain-SF and trypsin-SF. The MW distribution of bromelain-SF was relatively concentrated, with the highest content of high-molecular-weight segments, while the MW of Na_2_CO_3_-SF was widely distributed, mostly in low-molecular-weight regions. The above results showed that degumming with different proteases would degrade SF to different degrees, but all enzymes caused less degradation than did Na_2_CO_3_. Bromelain caused the least degradation of SF, whereas trypsin and papain caused relatively greater SF degradation.

In order to reduce damage to silk fibroin, several studies have optimized alkaline degumming conditions. At a lower temperature, lower Na_2_CO_3_ concentration or within a short time, degumming with Na_2_CO_3_ could also completely remove sericin and obtain high-molecular-weight silk fibroin with the MW ranging between 150 and 250 kDa [10], which was close to the MWs of subtilisin-SF, trypsin-SF, bromelain-SF and papain-SF (Figure 4 and Figure 5). However, alkali degumming will cause non-specific protein degradation. When not completely degummed, the alkaline solution will reach the internal fibroin at the part that has been completely degummed, leading to excessive degumming and a widespread distribution of the MW of silk fibroin. Enzymatic degumming is specific. Compared to alkali degumming, enzymatic degumming is milder and more uniform, resulting in a more concentrated distribution of the MW of silk fibroin [4,10,16,21,40,41,42], which is also verified in Figure 1B and Figure 6.

The viscous flow of polymers is the result of the displacement of the center of gravity of molecular chains along the flow direction and the mutual sliding between chains. The higher the MW is, the more segments a molecular chain contains, the more times the chain segment collaborative displacement needs to be completed to displace the center of gravity, and the greater the shear viscosity is [3]. At low shear rates (0.01–10 rad/s), macromolecules are in a highly entangled structure with high flow resistance; although the entangled structure can be destroyed, the speed of destruction is equal to the speed of formation, so the viscosity of each regenerated SF aqueous solution remains constant at the highest level. The results in Figure 7 show that bromelain-SF had the highest MW, Na_2_CO_3_-SF had the lowest MW, and there was no significant difference in MWs between subtilisin-SF and trypsin-SF, which were between those of bromelain-SF and papain-SF. The MWs of regenerated SF derived from degumming with the four neutral proteases were higher than that of SF derived from Na_2_CO_3_ degumming, which was consistent with the results of SDS–PAGE and GPC. When the shear rate increases, the entanglement structure is destroyed faster than it forms, which leads to a decrease in shear viscosity and the phenomenon of “shear thinning”. When the shear rate is increased to 1000 rad/s, the shear force promotes the aggregation of SF macromolecules [43], which ultimately precipitate. The spatial structure of SF is completely changed under the action of shear, and the solution viscosity reaches a constant minimum.

## 5. Conclusions

In this study, the degumming efficiency of several neutral proteases in silk degumming and the influence on the molecular weight of regenerated silk fibroin were investigated. The results indicated that when silk was treated with subtilisin, trypsin, bromelain and papain, sericin could be removed from silk almost completely, and the molecular weight of regenerated silk fibroin derived from neutral protease degumming was significantly higher than that of SF derived from Na_2_CO_3_ degumming. The thermal stability of bromelain-silk was the best, and the content of high-molecular-weight segments in bromelain-SF was the highest, indicating that bromelain caused the least damage to silk fibroin. Therefore, degumming with bromelain can not only fully remove sericin but also maintain the integrity of silk fibroin, which provides an efficient and environmentally friendly biological degumming method to prepare high-molecular-weight regenerated silk fibroin and contributes to improving the mechanical properties of SF-based materials and delaying their biodegradation rate. Regenerated SF derived from bromelain has broad application prospects in the field of tissue regeneration, especially in hard tissue.

## Figures and Tables

**Figure 1 polymers-15-03383-f001:**
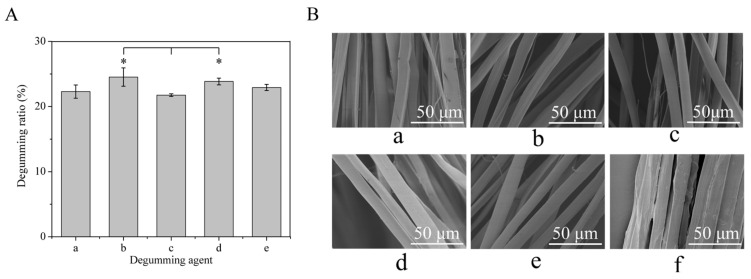
Degumming ratio of different degumming methods and the surface morphologies of degummed silk fibers. (**A**) Degumming ratio: (a) subtilisin; (b) trypsin; (c) bromelain; (d) papain; (e) Na_2_CO_3_. (**B**) SEM images: (a) subtilisin-silk; (b) trypsin-silk; (c) bromelain-silk; (d) papain-silk; (e) Na_2_CO_3_-silk; (f) raw silk. * Indicates significant differences compared to bromelain at *p* < 0.05.

**Figure 2 polymers-15-03383-f002:**
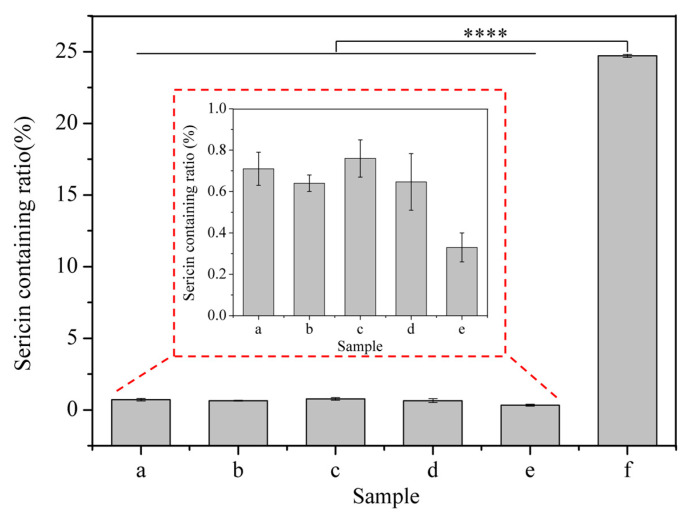
Figure **2.** Sericin containing ratio of degummed silk fibers and raw silk. (a) Subtilisin-silk; (b) trypsin-silk; (c) bromelain-silk; (d) papain-silk; (e) Na_2_CO_3_-silk; (f) raw silk. **** Indicates significant differences compared with raw silk at *p* < 0.0001.

**Figure 3 polymers-15-03383-f003:**
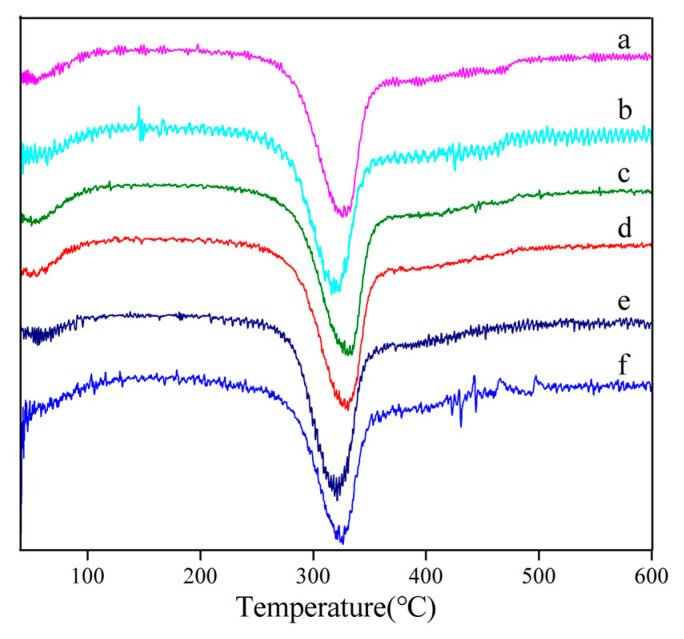
DTG curves of degummed silk fibers and raw silk. (a) Subtilisin-silk; (b) trypsin-silk; (c) bromelain-silk; (d) papain-silk; (e) Na_2_CO_3_-silk; (f) raw silk.

**Figure 4 polymers-15-03383-f004:**
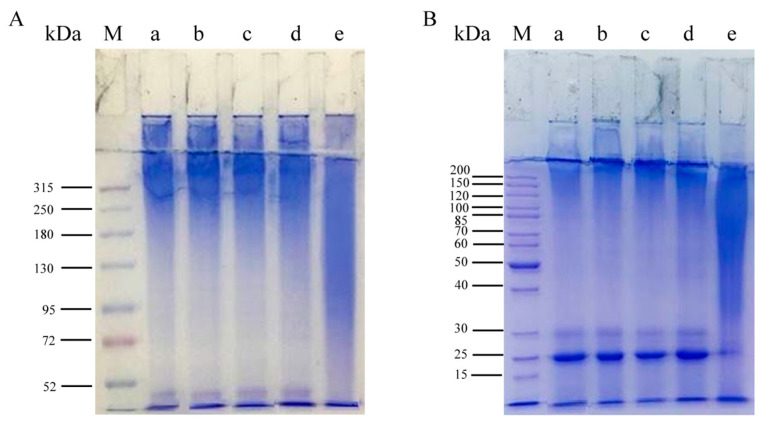
SDS–PAGE patterns of the regenerated silk fibroin derived from degumming with different proteases. Separating gel concentration: (**A**) 6%; (**B**) 10%. (M) Molecular weight markers; (a) subtilisin; (b) trypsin; (c) bromelain; (d) papain; (e) Na_2_CO_3_.

**Figure 5 polymers-15-03383-f005:**
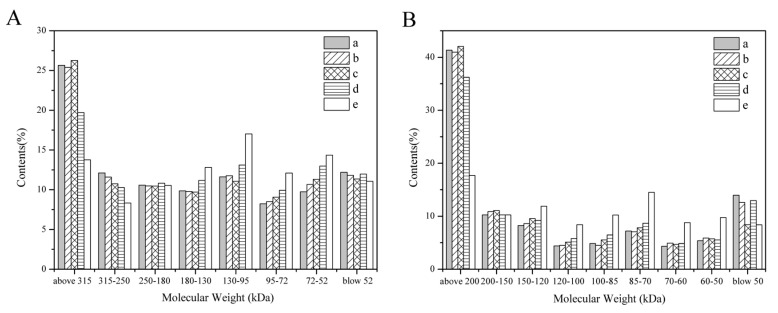
The molecular weight distribution of the silk fibroin derived from degumming with different proteases. (a) Subtilisin; (b) trypsin; (c) bromelain; (d) papain; (e) Na_2_CO_3_. Separating gel concentration: (**A**) 6%; (**B**) 10%.

**Figure 6 polymers-15-03383-f006:**
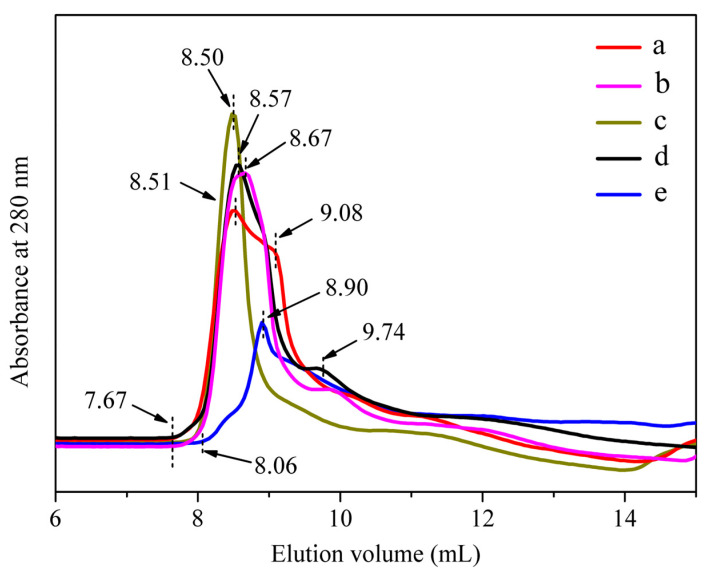
Chromatographic leaching curves of silk fibroin solution derived from degumming with different proteases. (a) Subtilisin; (b) trypsin; (c) bromelain; (d) papain; (e) Na_2_CO_3_.

**Figure 7 polymers-15-03383-f007:**
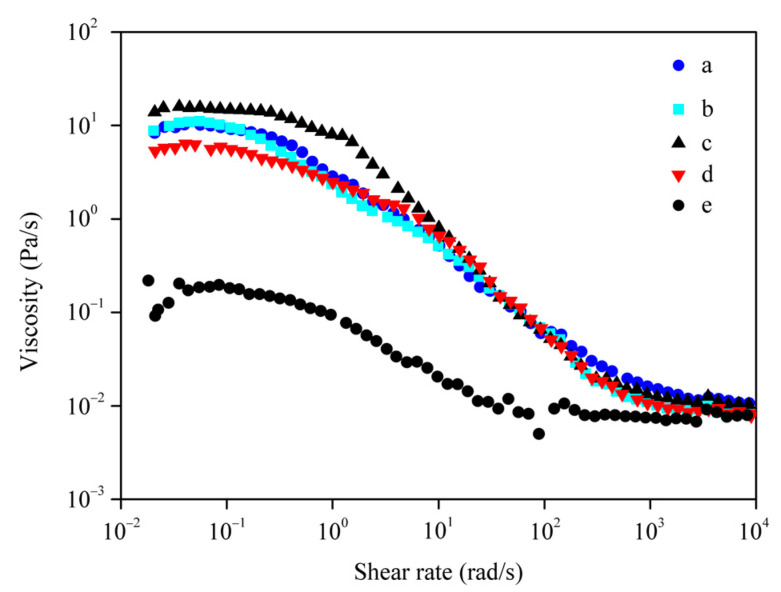
Diagram of viscosity changes with shear rate of silk fibroin solution derived from degumming with different proteases. (a) Subtilisin; (b) trypsin; (c) bromelain; (d) papain; (e) Na_2_CO_3_.

**Table 1 polymers-15-03383-t001:** The enzymatic degumming conditions.

Enzyme	Concentration (g/L)	pH	Temperature (°C)	Each Time (min)
Subtilisin	4	7	50	60
Trypsin	4	6.8	38	60
Bromelain	4	6.4	40	60
Papain	2	6.6	55	60
Na_2_CO_3_	0.5	/	100	30

**Table 2 polymers-15-03383-t002:** The amino acid composition (mol%) of raw silk and degummed silk fibers.

Amino Acids	Subtilisin	Trypsin	Bromelain	Papain	Na_2_CO_3_	Raw Silk
Asp	1.75 ± 0.04	1.76 ± 0.03	1.78 ± 0.01	1.73 ± 0.01	1.69 ± 0.01	3.94 ± 0.03
Thr	0.98 ± 0.02	0.91 ± 0.07	0.97 ± 0.01	0.89 ± 0.14	0.93 ± 0.01	2.36 ± 0.04
Ser	10.08 ± 0.18	11.91 ± 1.38	10.78 ± 0.15	10.92 ± 0.02	10.85 ± 0.08	14.67 ± 0.21
Glu	1.86 ± 0.06	1.01 ± 1.17	1.93 ± 0.17	2.03 ± 0.18	1.57 ± 0.01	2.36 ± 0.21
Gly	42.21 ± 0.08	41.33 ± 0.45	41.70 ± 0.76	41.78 ± 0.48	43.30 ± 0.52	37.89 ± 0.16
Ala	31.41 ± 0.47	31.73 ± 0.32	31.84 ± 0.15	31.13 ± 0.07	31.12 ± 0.27	27.19 ± 0.10
Val	2.29 ± 0.04	2.32 ± 0.10	2.36 ± 0.07	2.43 ± 0.23	2.22 ± 0.00	2.46 ± 0.03
Ile	0.56 ± 0.03	0.55 ± 0.04	0.60 ± 0.03	0.6 ± 0.08	0.56 ± 0.01	0.58 ± 0.02
Leu	0.54 ± 0.00	0.54 ± 0.05	0.54 ± 0.00	0.60 ± 0.08	0.51 ± 0.01	0.62 ± 0.04
Tyr	5.04 ± 0.16	4.91 ± 0.20	4.92 ± 0.01	5.17 ± 0.02	5.04 ± 0.12	4.63 ± 0.08
Phe	0.86 ± 0.06	0.80 ± 0.05	0.85 ± 0.06	0.88 ± 0.03	0.74 ± 0.04	0.71 ± 0.05
Lys	0.34 ± 0.04	0.34 ± 0.01	0.34 ± 0.04	0.34 ± 0.03	0.27 ± 0.04	0.71 ± 0.02
His	0.25 ± 0.03	0.24 ± 0.03	0.25 ± 0.03	0.25 ± 0.04	0.21 ± 0.01	0.36 ± 0.02
Arg	0.46 ± 0.01	0.46 ± 0.03	0.46 ± 0.01	0.44 ± 0.01	0.45 ± 0.01	0.85 ± 0.05
Pro	0.70 ± 0.16	0.64 ± 0.17	0.71 ± 0.13	0.79 ± 0.04	0.56 ± 0.01	0.67 ± 0.06
Total	100.00	100.00	100.00	100.00	100.00	100.00

## Data Availability

The data that support the findings of this study are available from the corresponding author upon reasonable request.

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
