# Peer review of "Comparative Study of the Preparation of High-Molecular-Weight Fibroin by Degumming Silk with Several Neutral Proteases"

_polymers, 2023, doi:10.3390/polym15163383_

Round 1

Reviewer 1 Report

Please consider the following when revising the manuscript:

1) Add statistical analysis information to the figures.

2) Add Na2CO3 degumming condition in Table 1.

3) Add data for raw silk in all Tables and Figures.

Good in general. Nedd improvement in a few cases.

Author Response

  1. Add statistical analysis information to the figures.

Response: Thank you so much for your comments. According to your comments, we added the statistical analysis information to the Figure 1(A) and Figure 2. The relevant descriptions have been added in page 5 line 189.

  1. Add Na2CO3 degumming condition in Table 1.

Response: According to your comments, we added Na2CO3-degumming condition in Table 1.

  1. Add data for raw silk in all Tables and Figures.

Response: According to your comments, we have added the data for raw silk in Figure 2 and Figure 3. The relevant descriptions have been added in page 5 line 212-213 and page 7 line 227-228.

Reviewer 2 Report

The article «Comparative Study of the Preparation of High-Molecular-Weight Fibroin by Degumming Silk with Several Neutral Proteases» is attracted to a relevant topic.

This study aimed to removing sericin from the periphery of silk without damage to silk fibroin. In this study, four neutral proteases, subtilisin, trypsin, bromelain and papain, were used to degum silk. The results indicated that all four neutral proteases could remove sericin from silk almost completely, and they caused less damage to silk fibroin fibers than Na2CO3 degumming. Thus, this study provides an efficient and environmentally friendly biological degumming method for obtaining high-molecular-weight silk fibroin.

The article is well structured, written in sufficient detail and logically. Introduction covers fundamental works in this direction and modern literary sources, but in my opinion, the conclusion should be expanded, adding key results by specifying key values and materials on the prospects and possible current directions of research in the field. This is my main note.

Minor remarks:

Line 74: “active center” should be replaced with “active site”.

The heading of Section 2 and the section two itself should be placed on the same page.

In Figure 4, the electrophoresis track labels at the top of the Figure are slightly shifted to the left.

Minor editing of English language required

Author Response

The article «Comparative Study of the Preparation of High-Molecular-Weight Fibroin by Degumming Silk with Several Neutral Proteases» is attracted to a relevant topic.

This study aimed to removing sericin from the periphery of silk without damage to silk fibroin. In this study, four neutral proteases, subtilisin, trypsin, bromelain and papain, were used to degum silk. The results indicated that all four neutral proteases could remove sericin from silk almost completely, and they caused less damage to silk fibroin fibers than Na2CO3 degumming. Thus, this study provides an efficient and environmentally friendly biological degumming method for obtaining high-molecular-weight silk fibroin.

The article is well structured, written in sufficient detail and logically. Introduction covers fundamental works in this direction and modern literary sources, but in my opinion, the conclusion should be expanded, adding key results by specifying key values and materials on the prospects and possible current directions of research in the field. This is my main note.

Response: Thank you so much for your comments. According to your comments, we expanded the conclusion in page 11-12 line 408-412.

  1. Line 74: “active center” should be replaced with “active site”.

Response: According to your comments, we have replaced the “active center” with “active site” in line 77.

  1. The heading of Section 2 and the section two itself should be placed on the same page.

Response: According to your comments, we adjusted the heading of Section 2 and the section two itself on the same page.

  1. In Figure 4, the electrophoresis track labels at the top of the Figure are slightly shifted to the left.

Response: According to your comments, we have shifted the electrophoresis track labels in Figure 4.

Reviewer 3 Report

The authors performed a comparative study on silk degumming with different enzymes using as a reference the degumming procedure with sodium carbonate. The novelty of the paper is not strong (several other papers performed the degumming with different enzymes) in particular the following paper "Degumming of silk with different protease enzymes" by Gulrajani et al. , and several other similar papers on the field. 

The main issue with this paper is the reference samples against which the other degumming were compares.  This reference is misleading in fact the author used a protocol harsher than the Rockwood protocol, demonstrating that the enzymes performed better. I would suggest a major revision due to the lack of a proper reference sample.

I would suggest a major revision.

Comments:

1)      The author should include the following reference: “A Design of Experiment Rational Optimization of the Degumming Process and Its Impact on the Silk Fibroin Properties” which describe an optimization of the alkali degumming demonstrating that a complete degumming is feasible even at lower temperature and lower alkali concentration. This paper should be used also to discuss the results obtained by the author comparing the Mw.

2)      The degumming procedure that was used for the reference is not the standard procedure that require a single bath of 30 min in boiling water with 0.05% Na2CO3 (Rockwood protocol). This protocol requires 1h less than your protocol. The author should at least prepare a reference sample with the Rockwood protocol and then do the comparison.

3)      The fiber diameters should be evaluated starting from the SEM images and the result should be coherent with the weight lost in the process.

4)      A single replica in paragraph 3.4 is not sufficient to understand if the results are significant. I would suggest repeating the process and perform the SDS-PAGE again mediating the results.

5)      The rheological measurement should also be repeated with more than one replica.

6)      Tensile measurements on the fibers are also a common method to understand how the degumming influenced the properties of the material. I would suggest to mechanically test the fibers.

The quality of the paper is good I would only suggest some check to correct typos and minor errors. 

Author Response

The authors performed a comparative study on silk degumming with different enzymes using as a reference the degumming procedure with sodium carbonate. The novelty of the paper is not strong (several other papers performed the degumming with different enzymes) in particular the following paper "Degumming of silk with different protease enzymes" by Gulrajani et al., and several other similar papers on the field. 

The main issue with this paper is the reference samples against which the other degumming were compares. This reference is misleading in fact the author used a protocol harsher than the Rockwood protocol, demonstrating that the enzymes performed better. I would suggest a major revision due to the lack of a proper reference sample.

I would suggest a major revision.

Response: Thank you so much for your comments. As the reviewer said, several other papers performed the degumming with different enzymes. In the paper “Degumming of silk with different protease enzymes” by Gulrajani et al., Degummase 1000L, Protosol, Trypsin, Alcalase, Protease A, Protease N, Pepsin and Protease M, eight proteases were used to degum silk. However, among these proteases, Degummase 1000L, Protosol, Trypsin and Alcalase worked under alkaline conditions, and Pepsin and Protease M worked under acid conditions. Both alkaline and acidic conditions will cause damage to silk fibroin during degumming, especially alkaline conditions, while neutral conditions cause less damage to silk fibroin. Therefore, in our study, we did not consider the proteases that work under acidic conditions and alkaline conditions, but mainly compared the influence of several proteases (subtilisin, trypsin, bromelain and papain) that work under neutral or near neutral conditions on the molecular weight of silk fibroin.

In Rockwood protocol, B. mori silk cocoons were degummed in a boiling bath of 2.12 g/L Na2CO3 solution for 30 min. In our study, the raw silk was boiled three times in 0.05% (0.5 g/L) Na2CO3 for 30 min each. The concentration of Na2CO3 in Rockwood protocol is higher. High concentration of Na2CO3 may cause significant damage to silk fibroin and researches have shown that the protocol in our study can completely remove sericin and causes less damage to silk fibroin (Lu S, et al. Mater. Today Commun., 2021, 27, 102369; Wang Y, et al. Front. Bioeng. Biotechnol., 2022, 9, 777320.). The control group of Na2CO3-degumming method in this study has been used by many researchers to extract silk fibroin and prepare silk fibroin-based materials (Zheng H, et al. ACS Appl. Mater. Interfaces, 2021, 40013-40031; Feng Y, et al. Carbohyd. Polym., 2019, 216,17-24.). Moreover, the Na2CO3-degumming method in this study has been incorporated into the national standard of the People's Republic of China as a standard degumming procedure for determining the sericin content in silk (GB/T 1798-2008, Testing method for raw silk). For these reasons, the method of degumming silk for 3 times (30 min each time) with 0.5 g/L Na2CO3 was selected as the control group in this study. We think it is reasonable to use this degumming method as the control group.

  1. The author should include the following reference: “A Design of Experiment Rational Optimization of the Degumming Process and Its Impact on the Silk Fibroin Properties” which describe an optimization of the alkali degumming demonstrating that a complete degumming is feasible even at lower temperature and lower alkali concentration. This paper should be used also to discuss the results obtained by the author comparing the Mw.

Response: According to your comments, we include the reference “A Design of Experiment Rational Optimization of the Degumming Process and Its Impact on the Silk Fibroin Properties” in our study. The relevant descriptions have been added in page 2 line 48-51. However, due to the different methods of dissolving silk fibroin fibers and the different methods of determining MW of silk fibroin, we did not discuss the results of the MW in the two papers.

  1. The degumming procedure that was used for the reference is not the standard procedure that require a single bath of 30 min in boiling water with 0.05% Na2CO3 (Rockwood protocol). This protocol requires 1h less than your protocol. The author should at least prepare a reference sample with the Rockwood protocol and then do the comparison.

Response: In Rockwood protocol, B. mori silk cocoons were degummed in a boiling bath of 2.12 g/L Na2CO3 solution for 30 min. In our study, the raw silk was boiled three times in 0.05% (0.5 g/L) Na2CO3 for 30 min each. The concentration of Na2CO3 in Rockwood protocol is higher. High concentration of Na2CO3 may cause significant damage to silk fibroin and researches have shown that the protocol in our study can completely remove sericin and causes less damage to silk fibroin (Lu S, et al. Mater. Today Commun., 2021, 27, 102369; Wang Y, et al. Front. Bioeng. Biotechnol., 2022, 9, 777320.). The control group of Na2CO3-degumming method in this study has been used by many researchers to extract silk fibroin and prepare silk fibroin-based materials (Zheng H, et al. ACS Appl. Mater. Interfaces, 2021, 40013-40031; Feng Y, et al. Carbohyd. Polym., 2019, 216,17-24.). Moreover, the Na2CO3-degumming method in this study has been incorporated into the national standard of the People's Republic of China as a standard degumming procedure for determining the sericin content in silk (GB/T 1798-2008, Testing method for raw silk). For these reasons, the method of degumming silk for 3 times (30 min each time) with 0.5 g/L Na2CO3 was selected as the control group in this study. We think it is reasonable to use this degumming method as the control group.

  1. The fiber diameters should be evaluated starting from the SEM images and the result should be coherent with the weight lost in the process.

Response: According to your comments, we evaluated the fiber diameters and did the statistical analysis. The diameters of subtilisin-silk, trypsin-silk, bromelain-silk, papain-silk, Na2CO3-silk and raw silk were 11.6 ± 1.3, 11.1 ± 1.6, 10.0 ± 1.0,       10.6 ± 2.3, 10.8 ± 1.7 and 12.8 ± 1.6 μm, respectively. The result indicated that the diameters of degummed silk fibers were smaller than that of raw silk and there was no obvious difference between the diameters of degummed silk fibers. The results of fiber diameters were included in Supplementary Materials (Figure S1), and the relevant descriptions have been added in page 5 line 199-200.

  1. A single replica in paragraph 3.4 is not sufficient to understand if the results are significant. I would suggest repeating the process and perform the SDS-PAGE again mediating the results.

Response: According to your comments, we repeated the process and perform the SDS-PAGE again and included the results in Annex I (Figure 1, Figure 2). The results of repeated experiments were similar to those in Figure 4 in the manuscript. In order to avoid the redundancy of data and images, only the image of one experiment was given as a representative of the experimental results in the manuscript.

  1. The rheological measurement should also be repeated with more than one replica.

Response: According to your comments, we repeated the rheological measurement several times and included the results in Annex I (Figure 3). The results of repeated experiments were similar to those in Figure 7 in the manuscript. In order to avoid the redundancy of data and images, only the image of one experiment was given as a representative of the experimental results in the manuscript.

  1. Tensile measurements on the fibers are also a common method to understand how the degumming influenced the properties of the material. I would suggest to mechanically test the fibers.

Response: As the reviewer said, tensile measurements on the fibers are also a common method to understand how the degumming influenced the properties of the material. However, in this study, we focused on the influence of degumming on the molecular weight of regenerated silk fibroin. We did not measure the tensile properties of the fibers.

Annex I

Figure 1. SDS‒PAGE patterns of the regenerated silk fibroin derived from different protease degumming. Separating gel concentration: (A) 6%, (B) 10%. (M) Molecular weight markers; (a) subtilisin; (b) bromelain; (c) trypsin; (d) papain; (e) Na2CO3.

Figure 2. The molecular weight distribution of the silk fibroin derived from degumming with different proteases. (a) Subtilisin; (b) bromelain; (c) trypsin; (d) papain; (e) Na2CO3. Separating gel concentration: (A) 6%, (B) 10%.

Figure 3. Diagram of viscosity changes with shear rate of silk fibroin solution derived from degumming with different proteases. (a) Subtilisin; (b) trypsin; (c) bromelain; (d) papain; (e) Na2CO3.

Round 2

Reviewer 3 Report

The authors performed a comparative study on silk degumming with different enzymes using as a reference the degumming procedure with sodium carbonate. The novelty of the paper is not strong as stated in my previous review. 

However the authors answered to almost all my comments. I have only a minor point: The reference I suggested demonstrates that alkaly degumming can produce high molecular weight silk fibroin, this should be discussed becouse is the main point discussed in your work. 

There are few minor errors that needs to be corrected. 

Author Response

Thank you again for your comments. According to your comments, we discussed the results of the MW in the two papers. The relevant descriptions have been added in page 11 line 379-390.